# Will groundwater-borne nutrients affect river eutrophication in the future? A multi-tracer study along the Elbe River

Julia Zill [1] , Axel Suckow [2] , Ulf Mallast [3] , Jürgen Sültenfuß [4] , Axel Schmidt [5] , Christian Siebert [1]

[1] Dept. Catchment Hydrology, Helmholtz Centre for Environmental Research (UFZ), Halle (Saale), 06120, Germany
[2] CSIRO Land and Water, Urrbrae (SA), 5064, Australia
[3] Dept. Monitoring- and Exploration Technologies, Helmholtz Centre for Environmental Research (UFZ), Leipzig, 04318, Germany
[4] Institute of Environmental Physics, University of Bremen, Otto-Hahn-Allee 1, 28359, Bremen, Germany
[5] Ref. G4 Radiology and Water Monitoring, Federal Institute of Hydrology (BfG), Koblenz 56068, Germany

*Correspondence to*: Julia Zill (julia.zill@ufz.de) and Christian Siebert (christian.siebert@ufz.de)

## Abstract

The Elbe River drains an intensely used agricultural area and cuts through a series of consolidated and unconsolidated aquifers with heterogeneous hydraulic properties and dimensions. For decades and as a result of overfertilization, particularly in the former GDR, the hosted groundwater transport nutrients into the river with serious implications for water quality and ecosystem health. As fertilization practices changed over time, nutrient loads in the groundwater recharge declined since the 1990s. This study investigates the residence time scales of groundwater along the river, as a measure to estimate the input periods of the associated nutrients entering the river using multi-environmental tracers ($^3$H/$^3$He, $SF_6$, CFCs, $^{14}$C). By applying lumped parameter models, we conclude that the average ages of groundwater range from a few up to 41 years, with infiltration occurring predominantly after 1985. Our results identify a young groundwater system with measurable denitrification and minimal to moderate admixtures of older water fractions clearly discernible with $^4$He. That indicates, the groundwater that was recharged during the GDR period (1945-1989) at maximum fertilizer application has already run off, the nutrient concentrations in the groundwater have peaked and may continue to decline in the coming decades. These results are crucial for informing river basin management strategies aimed at mitigating eutrophication and protecting aquatic ecosystems. It provides valuable insights into the temporal dynamics of groundwater contributions to surface waters and their regional implications for sustainable resource management.

## 1. Introduction

In Central Europe, as in most regions worldwide, groundwater is a significant source of both nutrients and contaminants entering riverine systems (Shishaye et al., 2021). A potential source is the intense application of N-fertilisers, including slurry, that leads to large inputs of nitrate through agricultural land into the groundwater (van der Grift et al., 2016; Craswell, 2021; Chen et al., 2023), which may discharge as base flow into surface waters. This enhances the vulnerability towards eutrophication (Brookfield et al., 2021; Zill et al., 2024) and, thus jeopardising ecosystem stability (Dodds, 2006)

downstream to the ocean. During the last decades, important catchment decisions significantly reduced
the application of liquid manure and mineral N-fertilisers in Germany and Europe: i.e. the 1989 political
revolution in Eastern Germany, the 1991 EU Council Directive to protect waters against pollution by
nitrate (91/676/EWG), the year 2000 implementation of the European Water Framework Directive
(2000/60/EG), the 2017 German Fertiliser Ordinance (DüV2017) and in its 2020 modified form
(DüV2020). Nevertheless, to assess the possible future trends of nutrient loads in rivers, it would be
beneficial to link the current nutrient concentrations (such as $NO_3$ and $PO_4$) in potentially contributing
aquifers to specific periods of past fertilizer use. Sanford & Pope (2013) and Martin et al. (2017)
highlight the significance of understanding these timescales, as it is difficult to establish and achieve
objectives for preserving surface water quality and evaluating the effects of groundwater-borne
nutrients on river systems without this knowledge. This understanding could facilitate the development
of regionally specific interventions and the implementation of effective management strategies to
reduce eutrophication and safeguard aquatic ecosystems.
Furthermore, the study of groundwater age distributions and mean residence times is valuable for
several other reasons (Dagan, 1986; Dagan & Nguyen, 1989), as it allows the validation of groundwater
models, ensuring their accuracy and reliability (Sanford, 2011; Schilling et al., 2019; Ying et al., 2023).
Understanding age distributions helps in quantifying groundwater infiltration and recharge rates, which
are essential for water resources management. Gilmore et al. (2016) emphasise the significance of
accurately determining groundwater flow paths from the source to the discharge areas, being important
in revealing the transport behaviour of both temporary and persistent contaminants. Investigating
groundwater age as tracer age provides a deeper understanding of flow processes within the aquifer,
contributing to more effective groundwater management practices. And, assessing tracer age is relevant
for determining the vulnerability of the aquifer and establishing protection zones, such as those for
drinking water (Molson & Frind, 2012) or groundwater-dependent areas in general (Isokangas et al.,

69     2017).

However, groundwater samples taken at one point represent an inextricable mix of water having
observed short and long flow paths and therefore containing fractions of very different ages. The age
distribution rather than the "mean groundwater age" is therefore the essential measure for precise root-
cause analysis and the assessment of the vulnerability of a groundwater body with regard to
contaminants or nutrients (McCallum et al., 2014; Suckow, 2014). Determining the age distribution as
a whole is, however, only possible in exceptional cases with a very long time series of tracer
measurements (Suckow & Gerber 2022). Therefore, instead of a single tracer (radioisotopes, stable
isotopes or hydrochemical parameters), a combination of natural geochemical or isotopic tracers must
be used and analysed. These multi-environmental tracer studies often cover different time scales
(Corcho Alvarado et al., 2007; Mayer et al., 2014; Purtschert et al., 2023). Younger fractions can be
determined using $^3$H and tritiogenic $^3$He$_{trit}$ (Sültenfuß & Massmann, 2004; Desens et al., 2023; Wang et
al., 2023), anthropogenic gases like SF$_6$, CFC-11, CFC-12 (Daughney et al., 2010; Okofo et al., 2022)
and $^{85}$Kr (Cook & Solomon, 1997; Kagabu et al., 2017). Older water fractions are estimated by $^{14}$C
(Maloszewski & Zuber, 1991; Iverach et al., 2017), $^{36}$Cl (Wilske et al., 2019; Purtschert et al., 2023) or
$^4$He (Matsumoto et al., 2018). Even with multi-tracer applications the details of the age distributions
can only be pinpointed with a high level of uncertainty (McCallum et al., 2014), but nevertheless, this
remains the state-of-the art approach. Only knowledge of the different groundwater age fractions and
their spatial distribution in the subsurface offers the possibility of understanding the vulnerability of the
aquifer and its negative influence on surface waters through possible groundwater - surface water
interactions. To obtain mean residence times from tracer measurements, lumped parameters models
(LPM), which are detailed in Maloszewski & Zuber (1996, 2008) and Suckow (2014) are powerful
tools. In essence, Lumped Parameter Models (LPMs) are analytical solutions characterized by a limited
number of key parameters and a simplified aquifer structure. They employ a mathematical age
distribution to compute Mean Residence Times (MRTs) based on sampled tracer concentrations.
All nutrients in the aquifer are subject to dispersion and diffusion effects. With regard to the entry paths
of N and P species into watercourses, their different physico-chemical characteristics must be taken into
account. In contrast to phosphate, nitrate does not adsorb significantly to minerals and soil particles,
which results in direct surface runoff into the receiving waters. These only reach the river systems with
a corresponding time delay (legacy effect - Lautz et al., 2020; Martin et al., 2021; Shishaye et al., 2021),
which makes it difficult to predict nutrient concentrations (Kunkel & Wendland, 2006; Martin et al.,
2017). The calibration and validation of process-based models is possible for small-scale (a few km²)
applications, but there is still a need for research into their transferability to large-scale issues (>100
km²) (Briggs & Hare, 2018). Due to the extreme chemical heterogeneity at the interfaces of groundwater
and surface water, in connection with seasonally variable parameters like water level, oxygen or nutrient
concentrations, a generalised prediction of the influence of groundwater on ecosystems is not possible.
For this, catchment-specific regional studies must be used and limiting factors such as usable pore
volume, hydraulic connectivity, temperature, organic matter and riverine nutrient control mechanisms
must be analysed.
This study aims to estimate time scales of groundwater movement and nutrient transport within several
different aquifers along an almost 450 km stretch of the Elbe River in Central Germany. Earlier studies
revealed the accompanying groundwaters are sources of nutrients in the Elbe River, affecting the food
web and eutrophication of the river (Zill et al., 2023; 2024) and eventually the ocean (Kamjunke et al.,

112 2023).



## 2. Study Area


The studied groundwater bodies are located along the 450 km long stretch of the German Elbe River from Schöna at the Czech border to Wittenberge. The Elbe River is one of the largest rivers in Europe and shows strong eutrophication effects (Hardenbicker et al., 2016), being the major reason to downgrade its ecological state to 'moderate/unsatisfactory', according to the European water framework directive (UBA, 2022). The impact of groundwater-borne nutrients to the benthic and pelagic eutrophication of the Elbe River were studied by Zill et al. (2024) and show that groundwater contributes significantly to pelagic eutrophication and particularly under low flow conditions. Benthic eutrophication and the algae community are mainly dependent on the season and subsequently influenced by the aquifer type and groundwater discharge.

Zill et al. (2023) investigated the temporal and spatial interactions between groundwater and surface water along the Elbe River. Their findings reveal distinct differences in groundwater fluxes and resulting water quality. In the headwater regions, pronounced topography forces fast flow that leads to high groundwater discharge with low nutrient loads, as agricultural activity in these upstream catchments is minimal. In contrast, downstream lowland areas (Fig. 1B) show low natural groundwater inflow due to missing topographical gradients, while fertile soils lead to intense agriculture. The combination of both requires artificial dewatering of the fields by dikes and subsurface drainage systems, being both preferential channels to transport nutrient-rich soil water into the Elbe River. Their effect even increases during droughts, when hydraulic gradients are higher.

The hydrogeological situation in the study area is divided into two main settings (Fig. 1B). Within the first 100 stream km, the Elbe cuts through the hard rocks of the Saxonian Cretaceous, comprising the Elbtal group with coastal sandstones. Afterwards, the river runs through the mostly unconsolidated Pleistocene aquifers of the Middle German Lowland, which were deposited during glacial and interglacial periods (IKSE, 2005; Wilmsen & Niebuhr, 2014).

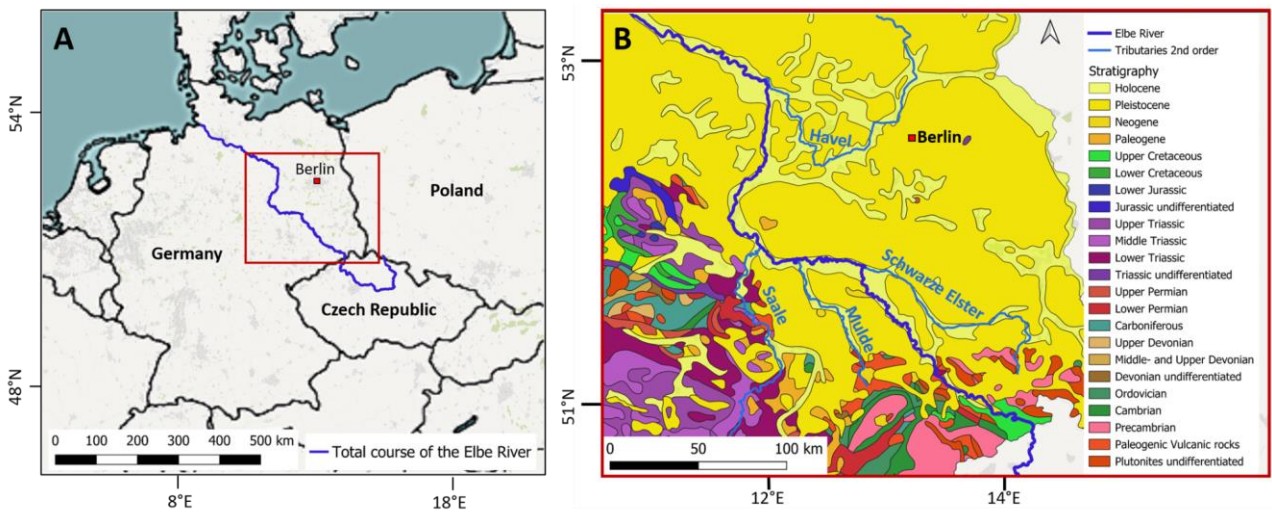

140

Figure 1. Elbe River study area: A) Location of the Elbe River in Central Europe. B) Stratigraphic map of the study area. Base Maps derived from QGIS desktop and stratigraphic map derived from the geological map of Germany (GK2750) from the Federal Institute for Geosciences and Natural Resources (BGR), Hannover.

The Pleistocene aquifers have a spatially variable permeability and a silicate, partly silicate-organic rock characteristic (Eissmann, 2002). The youngest deposits comprise an aquifer complex from Weichsel and Saale glacial period and are composed of ca. 20 m thick permeable fluvial sands and gravels. They are underlain by about 50 m of less permeable silty glauconite sands from the Elster glacial period. All the groundwater hosted in these formations drains towards the Elbe (Ad-Hoc-AG Hydrogeologie, 2016). The Cretaceous hard rocks consist of fissured and low to moderate hydraulic permeable sandstones, claystones, siltstones and bioturbated marlstones (Wilmsen & Niebuhr, 2014), forming an ca. 300 m thick complex of five aquifers, which predominantly drain towards the Elbe valley. In the Elbe valley and inside side valleys, groundwater tables are shallow, leading to a high vulnerability of the groundwater to contamination.

## 3. Data and Methods

To determine the age scales of groundwater and of the carried nutrients, which flow into the Elbe River, all the groundwater bodies which most probably contribute were identified using hydrogeological and contourline maps. Groundwater was sampled from these aquifers during two sampling campaigns in 2020 and 2022 by pumping from observation wells that (i) tap the target aquifers and are (ii) preferably located in close proximity (within a maximum of 10 km) to the Elbe Valley. The groundwater has been analysed for environmental age tracers covering a wide range of age spectra: $^{3}H/^{3}He$, $SF_6$, CFCs, and $^{14}C$. Subsequently, the lumped parameter model LUMPY (Suckow, 2012) was applied, to estimate time scales of groundwater ages based on a convolution integral including atmospheric input functions, borehole parameters such as depth and screened sections and conceptual flow models (exponential models (EM), dispersion models (DM), and piston flow models (PFM)). They represent the individual aquifer types and regional conditions. By visual comparison, we selected the best model for our data set.

### 3.1 Fundamentals of the methods

Tritium ($^{3}H$) is a radioactive isotope of hydrogen with a half-life of 12.32 years, naturally formed primarily in the atmosphere through spallation and neutron reactions of cosmic radiation with the atmosphere (Unterweger & Lucas 2000). Its concentration is measured in activity units (Bq/l). However, in environmental sciences, tritium concentrations are expressed in Tritium Units (TU). The conversion is 1 Bq/l = 8.47 TU (IAEA, 1992). In the hydrosphere, tritium becomes part of water

molecules and reaches the Earth via precipitation. Atmospheric nuclear tests in the late 1950s and 1960s caused a dramatic increase in tritium concentrations, raising them far above the natural background level of ~5 TU in the atmosphere (Clark & Fritz, 2013). Since the 1963 Nuclear Test Ban Treaty, atmospheric tritium levels have declined due to radioactive decay and dilution in the oceans. Today, they reach approximately 8 TU in the Northern Hemisphere, with slightly higher levels during the early summer months (Schmidt et al., 2020).

The tritium-helium dating method, developed in the 1980s (Poreda et al., 1988; Schlosser et al., 1988), uses tritium and its decay product tritiogenic helium-3 ($^3He_{trit}$), to estimate groundwater age. This method assumes there is no other source of $^3He_{trit}$ than the decay of $^3H$, while $^3He_{trit}$ needs to be separated from other $^3He$ sources. These are identified by measurements of $^4He$, which is produced from the decay of U and Th in the aquifer, and Ne (Schlosser et al., 1988). The sum of $^3H$ and $^3He_{trit}$ reflects the initial $^3H$ concentration at the time of water infiltration. Only in the absence of mixing and dispersion the so-called tritium-helium age indicates the advective groundwater velocity. Mixing or dispersion would bias results towards ages with higher tracer concentrations (Schlosser et al., 1988; Schlosser & Winckler, 2002). The tritium-helium age can be determined by the following equation:

$$\tau_{trit} = \frac{1}{\lambda_{trit}} \cdot ln \left(1 + \frac{3He_{trit}}{3_H}\right) \tag{1}$$

With $\tau_{trit}$ = tritium-helium age [a], $\lambda_{trit}$ = tritium decay constant [1/a], $^3He_{trit}$ = tritiogenic helium-3 concentration [TU], $^3H$ = tritium concentration [TU]. The concentration of $^3He_{trit}$ is determined from the helium and neon concentration on each sample using the formulas in Schlosser et al. (1988) and using a conversion factor of 1 cc(STP)/g = $4.019 \cdot 10^{14}$ TU.

Radiocarbon ($^{14}C$), with a half-life of $5,730 \pm 40$ years (Godwin, 1962), is used to estimate groundwater age by comparing the $^{14}C$ concentration of dissolved inorganic carbon (DIC) at recharge with current levels. Two main challenges include estimating the initial $^{14}C$ concentration at recharge and accounting for sub-surface dilution by $^{14}C$-free DIC from geochemical reactions during flow (Wigley et al., 1978; Plummer & Glynn, 2013). Atmospheric nuclear tests in the 1960s significantly increased $^{14}C$ levels, complicating age estimates for groundwater recharged before this period (Levin & Kromer, 1997). For dating, the initial $^{14}C$ concentration is typically assumed to be 100 pMC, reflecting pre-nuclear testing atmospheric $CO_2$ levels (Pearson & White, 1967). DIC in groundwater primarily originates from (i) soil $CO_2$ (high in $^{14}C$), where the $CO_2$ is derived from root respiration and organic material turnover, rather than direct atmospheric $CO_2$ and (ii) carbonate dissolution, where carbonates usually being low in $^{14}C$, except recently formed travertine (Geyh, 1970). Variations in atmospheric $^{14}C$, influenced by the earth's magnetic field, solar activity, and climate, can slightly alter age estimates, though these effects are minor compared to uncertainties from geochemical mixing of $CO_2$ and $CO_3$ into DIC (Pearson & Qua, 1993).

Chlorofluorocarbons (CFCs) and Sulphur hexafluoride (SF$_6$) are anthropogenic gaseous pollutants,
released into the atmosphere globally since the 1950s from industrial products and processes and
enriched in the atmosphere over time. It is assumed that the accumulation of these gases in the
atmosphere is in equilibrium with the concentration in surface waters at a given temperature and
pressure. Their respective atmospheric concentrations are well documented over time, providing a
continuous input function for groundwater dating, which is generally not the case for other gases. CFCs
were introduced in 1930 for applications such as refrigeration, air conditioning, aerosol propellants, and
plastic foams (IAEA, 2006). Their atmospheric concentration began decreasing only with the Montreal
Protocol in 1987, which restricted CFC use to protect the ozone layer. In contrast, sulphur hexafluoride
introduced in 1953 for use in gas-filled electrical switches, has seen a steady increase in the atmosphere
until the present. Its low solubility in water and high stability in soils make it an effective tracer for
dating young groundwater (Mroczek, 1997; Busenberg & Plummer, 2000). Groundwater dating with
CFCs and SF$_6$ depends on processes such as sorption, dispersion, and microbial degradation. SF$_6$ is due
to its resistance to degradation and limited affinity to sorption particularly reliable (Plummer &
Busenberg, 2000). The method assumes equilibrium governed by Henry's Law of solubility, with key
assumptions for recharge temperature, excess air entrainment, and unsaturated zone thickness
(Aeschbach-Hertig et al., 1999; IAEA, 2006).

3.2 Groundwater sampling

During spring 2020 and 2022, 63 boreholes and springs were sampled for groundwater from the shallow
aquifers that are in hydraulic connection with the Elbe River. The selected wells are located along 450
km of the river's course (Fig. 2). In the Saxonian Sandstone Mountain aquifers, where wells were
unavailable, samples were taken from springs. All sampling locations and analytical results are listed
in detail in the supplementary Tables S1 and S2. Groundwater was extracted using a Grundfos MP1
submersible pump (Eijkelkamp, Netherlands). Samples were collected after the standing water in the
boreholes was replaced three times and on-site parameters (pH, temperature, Eh and EC; measured with
multiparameter WTW 350i (WTW, Germany)) stabilised.

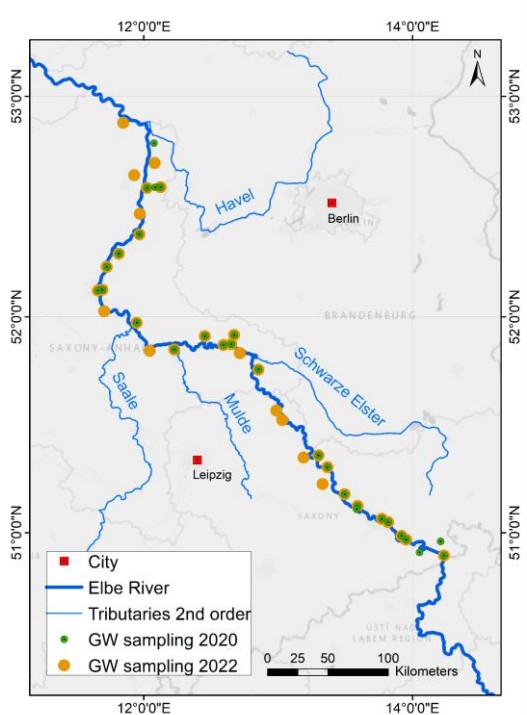

Figure 2: Map of all groundwater sampling locations from 2020 (green) and 2022 (orange) along the German Elbe
River.

Samples for $^3$H were filled in clean 500 ml HDPE containers. Duplicate samples for $^3$He were filled
carefully without air inclusions into 40 ml copper pipes of 1 m length, which were closed under pressure.
The samples for CFC-11, CFC-12 and SF$_6$ were taken by filling a 10 l HDPE container with the water
sample by overflow principle. Afterwards the container was closed without air inclusion. From below,
purified N$_2$-gas (Linde, Germany) was injected into the system to replace about 200 ml of water and to
generate a N$_2$-headspace. The closed system was then shaken for a minimum of 20 min to ensure the
degassing of water and equilibration with the headspace. Afterwards, parts of the headspace were
extracted at field temperature and injected into 20 ml glass vials, which have been evacuated and N$_2$-
flushed prior to sampling. Samples for $\delta^{13}$C/$\delta^{14}$C were filled in clean 60 l HDPE containers to ensure
sufficient amounts of DIC, while samples for $\delta^2$H/$\delta^{18}$O on water and $\delta^{15}$N/$\delta^{18}$O on dissolved NO$_3$ were
sterile filtered applying 0.22 µm filter (Sartobran, Sartorius, Germany) and filled into pre-cleaned
HDPE containers. All samples were stored dark and cool until laboratory analysis.
3.3 Analytical methods

The preparation of water samples for tritium analysis follows the method outlined by Schmidt et al.
(2020). This involves the electrolytic enrichment of a 300 - 500 ml water sample, followed by
measurement using liquid scintillation counting. The technique achieves a detection limit of 0.5 TU,
with a relative standard error of 10 % for $^3$H concentrations above 0.5 TU. All analyses were performed
using different liquid scintillation counters (TriCarb 3180, Quantulus GCT 6220, PerkinElmer,

Germany; Hidex 300 SL, Hidex Oy, Finland). The preparation and measurement of helium isotopes and neon in water samples follow the procedure described by Sültenfuß et al. (2009). In summary, gas is extracted from water contained in copper tubes and transferred into glass ampoules. For measurement, an ampoule is opened in a high-vacuum system, and the gases are transferred to low-temperature traps. Using a cryo trap at 25 K, He and Ne are separated from other gases. A fraction of the He-Ne mixture is analysed for $^4$He, $^{20}$Ne, and $^{22}$Ne using a quadrupole mass spectrometer (QMG 112, Pfeiffer Vacuum, Germany). The remaining gases are adsorbed to activated carbon in a low-temperature trap at 10 K, and after heating to 45 K, only He desorbs. This fraction is then analysed for $^3$He and $^4$He in a sector field mass spectrometer (MAP 215-50). The system was calibrated with atmospheric air, achieving a measurement accuracy of 0.4 % for $^3$He/$^4$He and $^4$He/$^{20}$Ne ratios, and 0.7 % for isotope concentrations ($1\sigma$ confidence interval; Sültenfuß et al., 2009).

The preparation and measurement of anthropogenic gases, including CFC-11, CFC-12, and SF$_6$, follow a method adapted from Busenberg and Plummer (2008). In this procedure, septum glass tubes are connected to a Rheodyne 6-way valve (Supleco, USA) with two operational positions: "load" (position I) and "inject" (position II). In the "load" position, the sample is directed into a stainless-steel trap (WICOM, Germany) filled with Porapak T and cooled to -70 °C. The trap is flushed with nitrogen at 50 ml/min, then heated to 95 °C to release the tracers. When the valve is switched to the "inject" position, the sample is directed into a gas chromatograph (GC-2014, Shimadzu, Japan) equipped with an electron capture detector (ECD). Helium is used as the carrier gas at a constant flow rate of 30 ml/min. The GC system includes a pre-column (1 m; heated to 80 °C) and main column (3 m; heated to 180 °C), both filled with HayeSep Q. Specific elution times are observed for SF$_6$, CFC- 12, and CFC- 11 at 4.5, 17.5, and 31.1 min, respectively. The ECD operates at 300 °C to ensure precise detection. The system was calibrated with gas standards from the University of Bremen, achieving an analytical uncertainty of ~2 % for SF$_6$ and CFCs.

The radiocarbon analyses were performed by liquid scintillation counting (Tri-Carb 3770 TR/SL, Perkin Elmer). First, BaCl$_2$ was added to the alkalized water samples to precipitate DIC as BaCO$_3$. Afterwards, precipitated carbon is transferred into CO$_2$ and further transformed into lithium carbide, acetylene and finally into benzene. NBS oxalic acids (USA) and ANU sucrose (Australia) were used as standards (Trettin et al., 2006). For $\underline{\delta^{13}C}$, part of the formed CO$_2$ was captured separately and kept for stable carbon isotope analysis in an isotope ratio mass spectrometer (Delta plus XL, Thermo Quest, Germany). As an internal standard, BaCO$_3$ was used which was calibrated on the international V-PDB (Vienna Pee Dee Belemnite) standard.

The stable isotopes of water ($\delta^2$H and $\delta^{18}$O) were measured by laser cavity ring-down spectroscopy (CRDS) using a Picarro L2120-1 (Picarro, USA). The analytical precision is $\pm0.1$ ‰ for $\delta^{18}$O and

±0.8 ‰ for $\delta^2H$ and results are reported relative to Vienna Standard Mean Ocean Water (V-SMOW).
Nitrate was measured with ion chromatography (Dionex ICS-2000, Thermo Scientific, Germany) and
the $\delta^{15}N$ and $\delta^{18}O$ signatures of dissolved $NO_3$ were determined with a denitrifier method using bacteria
strains of *Pseudomonas chlororaphis* (Sigman et al., 2001). Using a GasBench II with an isotope ratio
mass spectrometer (DELTA V Plus, Thermo Scientific, Germany), the resulting $N_2O$ gas from
microbial production was measured. The analytical precision is ±0.4 ‰ for $\delta^{15}N$ and ±0.8 ‰ for $\delta^{18}O$.
## 3.4 Modelling
Lumped parameter models (LPMs) are predefined analytical solutions for simplified flow systems that
conceptualise the entire aquifer as a spatially zero-dimensional system. The theoretical basis of LPMs
is described in detail by Maloszewski & Zuber (1982), Han & Maloszewski (2006), and the IAEA
(2006). Mathematically, LPMs are represented by a convolution integral, which integrates the time-
dependent input function using the age distribution as a weighting function. The concentration of a
certain substance, the „tracer", in groundwater sampled at a well or spring can be described by this
convolution integral, while the fitting parameter is the mean residence time (MRT):

$$C_{out}(t) = \int_0^\infty C_{in}(t') \cdot g(t-t') \cdot e^{(-\lambda(t-t'))} dt' \qquad (2)$$

With $C_{out}$ = output concentration, $C_{in}(t')$ = time dependent sampled input concentration, t = time, t' =
transit time or input time, g(t') = the age distribution, a weighting function that is normalised to 1. The
MRT and in some cases a secondary parameter such as the Péclet number characterise the shape of
the age distribution g(t').

In our study the LPM LUMPY was used (Suckow, 2012), which is part of the graphical user interface
of the LabData Database and Laboratory Management System from Suckow and Dumke (2001). The
algorithm of LabData allows the modelling of several tracers simultaneously, while the flow within the
aquifers were conceptualized by an exponential model (EM), a piston flow model (PFM) and a
dispersion model (DM). The LPM lines in the following figures are for orientation purposes only. A
PFM is conceptually the simplest of all approaches and describes a flow path in the aquifer without any
mixing, diffusion or dispersion. The resulting age distribution therefore contains exactly one age, so
that for this model the MRT and the idealised age are the same. An EM describes a completely mixed
aquifer (Eriksson, 1971) in a homogenous, unconfined system. The vertical zonation of the groundwater
age implies young ages at the groundwater table, increasing to infinity at the aquifer's base (Vogel,
1970; Appelo & Postma, 1996). A DM is characterised by the MRT and the Péclet number *Pe*, which
relates the magnitudes of dispersion and advection and is defined according to equation 3:

$$Pe = \frac{l \cdot v}{D} \qquad\qquad\qquad (3)$$

With l = characteristic length of the flow path [m], v = advective transport [m/s], D = dispersive
transport [m²/s].
Using a small Péclet number such as 4, results from a DM are very similar to an EM.

## 4. Results


### 4.1 Stable isotopes


The stable isotopic signatures ($\delta^2H$ and $\delta^{18}O$) of the sampled groundwater plot along the Global
Meteoric Water Line (GMWL: $\delta D = 7.9 \cdot \delta^{18}O + 9$ (Gat, 2001)), indicating low evaporation prior to
infiltration (Fig. 3A). The signatures range from -67 ‰ to -44 ‰ for $\delta^2H$ and from -9.5 ‰ to -5.5 ‰
for $\delta^{18}O$. Taking water sample #12 and calculating a theoretical evaporation line (at 60 % humidity and
20 °C), only samples #24 and #28 were exposed to higher evaporation (4.5 % and 8 %, respectively)
compared to all others. The two springs in Saxony have isotopic signatures slightly above the GMWL.
The $\delta^{13}C$ signatures of the sampled groundwater range from -23 ‰ to -12 ‰ V-PDB (Fig. 3B). This
mass range corresponds to carbon sources primarily derived from root respiration of C3 plants and
dissolution of freshwater carbonates (Meier-Augenstein, 1999). Additionally, distinct clusters (covering
over 80 % of all corresponding samples) of carbon species are discernible, with varying initial
concentrations of DIC detected among the samples. DIC concentrations range from 0 to 6.7 mMol/l,
with sample #3 showing the highest value of 6.7 mMol/l. The $\delta^{15}N$ in nitrate reveals significant
enrichment, with $\delta^{15}N$ values around +8 ‰ and $\delta^{18}O(NO_3)$ values approximately +2 ‰ V-SMOW (Fig.
3C), with sources of nitrate partly from soil organic matter (Fig. 4D). Three groundwater samples from
the Cretaceous aquifers show sources of mineralised fertiliser or manure (Fig. 4D, orange and green
rectangle). No consistent nitrate enrichment trend is evident, and a simple Rayleigh fractionation
calculation indicates that most samples observed less than 70 % denitrification (30 % residual) (Fig. 3C
and D, blue line) suggesting low to moderate denitrification rates within the different aquifer systems.
However, samples #7, #10, #13, #20, #22, #29 show high denitrification rates over 75 % with $NO_3$
concentrations close to the detection limit (see S1 and S2).

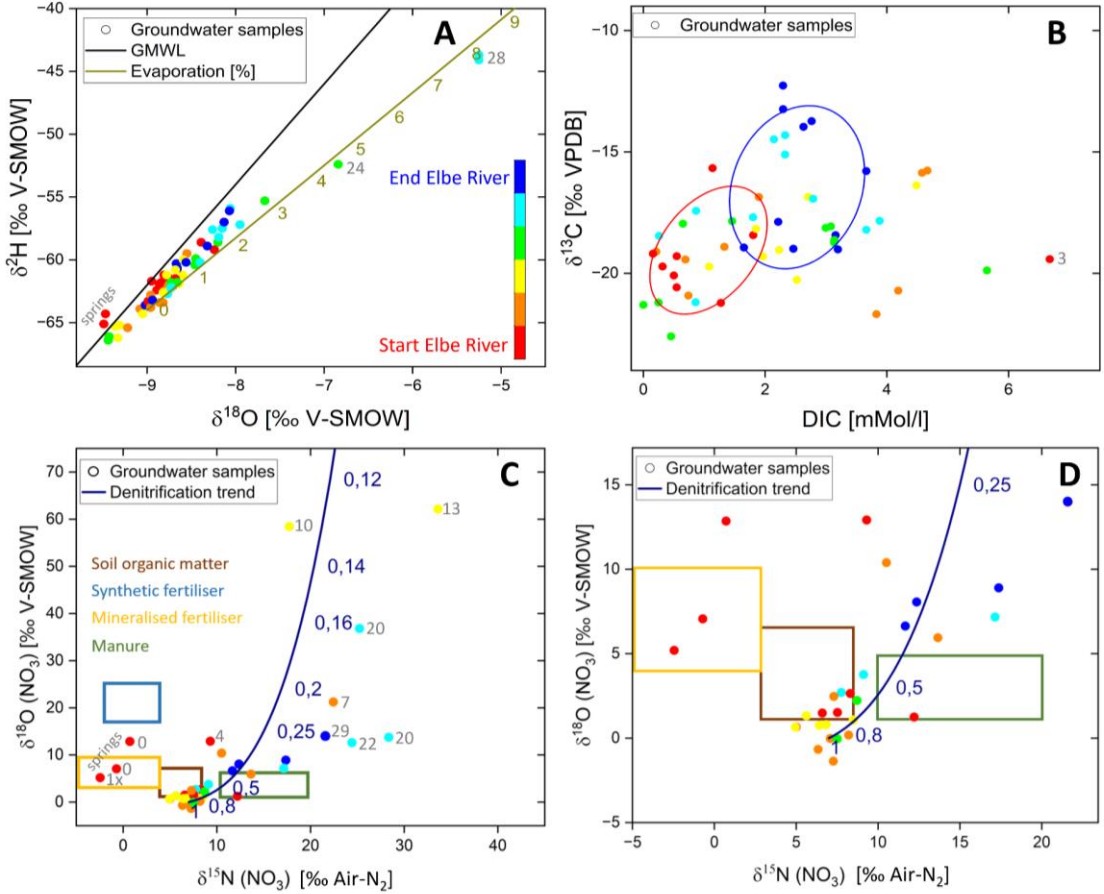

Figure 3. **A)** shows $\delta^2H$ versus $\delta^{18}O$ signatures in the sampled groundwaters. The khaki evaporation line indicates an accumulation of heavier isotopes through evaporation along the line. Numbers show evaporation rates in %. For better spatial orientation: Start and end of the Elbe River stands for groundwater samples which were taken from aquifers located along the start (warm colours) or end (cold colours) of the studied Elbe section, respectively. This color code was applied in figures 3 A to D. **B)** shows $\delta^{13}C$ signatures of DIC in relation to DIC in the sampled groundwater along the Elbe River. **C)** shows $\delta^{18}O$ versus $\delta^{15}N$ signatures of nitrate in the sampled groundwaters. The dark blue line shows Rayleigh fractionation of nitrate isotopes where the dark blue numbers present the non-fractionated part (remaining fraction of the initial $NO_3$ pool). Coloured frames show different nitrate sources according to Clark & Fritz (2013) of soil organic matter (brown), synthetic and mineralised fertiliser (blue and yellow, respectively) and manure (green). **D)** shows a zoom into figure 3C with the same nitrate sources as in figure 3C.

## 4.2. Environmental age tracers

Representative results from lumped parameter modelling are shown in Figures 4A to F.

In Figure 4A, the initial tritium concentration ($^3H + {}^3He$) is plotted against the infiltration year, suggesting mean residence times between 0 and 20 years, when using a dispersion model (DM) with a Péclet number of 4. Samples (e.g., #17 and #27) with concentrations below the levels found in the models (green and orange line, Fig. 4A) are indicative for a mixture with older, tritium-free groundwater that was recharged before the bomb tests in the 60s. Sample #27, in particular, is tritium-free water with

no detectable proportion of younger recharge and a major composition of an old water fraction (>60y). Conversely, samples with concentrations exceeding the precipitation level (e.g., #19, #31, #32) suggest the influence of water sources with elevated tritium levels, such as the cooling water release from the Czech nuclear power plant Temelín (Hanslík et al., 2017; Zill et al., 2023). Bank-filtrate could enhance $^3$H concentrations in groundwater as in sample #19, which is located along a stretch of the Elbe, where influent conditions prevail at least during times of low flow (Zill et al., 2023). This may exhibit anomalously high $^3$He values, accompanied by moderate $^3$H concentrations of 5.7 TU. The $^3$H concentrations in the collected groundwater samples are on average 5 TU and the corresponding $^3$H/$^3$He ages indicate residence times ranging from 2 to 52 years (Fig. 5). $^3$H/$^3$He ages exceeding 45 years are considered unreliable as their calculations are based on the detection limit in tritium (#27) or influenced by admixtures from the tritium peak and resulting high $^3$He (#19) concentrations (Fig. 4C). These $^3$H/$^3$He ages are therefore labelled as > 45 years in figure 5.

Even if a high proportion of young water is generally evident, tracers for older water such as $^4$He (Fig. 4E and F, discussed below) or radiocarbon (Fig. 4B) indicate an admixture of old water components for some samples (e.g. #0, #18, #24, #27, #28). Comparing young water fractions represented by $^3$H, with old water fractions represented by $^{14}$C, show noticeable deviations of groundwater samples from modelled graphs like an EM or a PFM (Fig. 4B). Most of the variation along the $^{14}$C axis in Fig. 4B can be explained by geochemical reactions of the DIC under open or closed system conditions, diluting the atmospheric $^{14}$C signal with $^{14}$C-free sedimentary carbon. However, springwater samples (Fig. 4B, red points) consistently appear as outliers. Sample #0 with 11 pMC shows a major contribution from old water fractions, while #2 with 117 pMC shows an infiltration about the time of nuclear bomb tests in the 1960s (as 100 pMC is by definition the year 1950). The contaminated sample #1 that can not be shown in the figures is the Mockenthal spring, which shows $^{14}$C values of 373 pMC and $^3$H values of 88 TU and 102 TU in 2020 and 2022, respectively. This is most probably a result of anthropogenic contamination by legacies from an abandoned military base or from the former repository of the GDR period in the catchment and therefore excluded from the analyses. A cluster of samples around 4 TU and 60 pMC indicates a multi-component system with mixtures of younger and older water fractions. Some of these samples (e.g. #26, #27, #28) are not visible in Figure 4D as it displays only young water fractions.

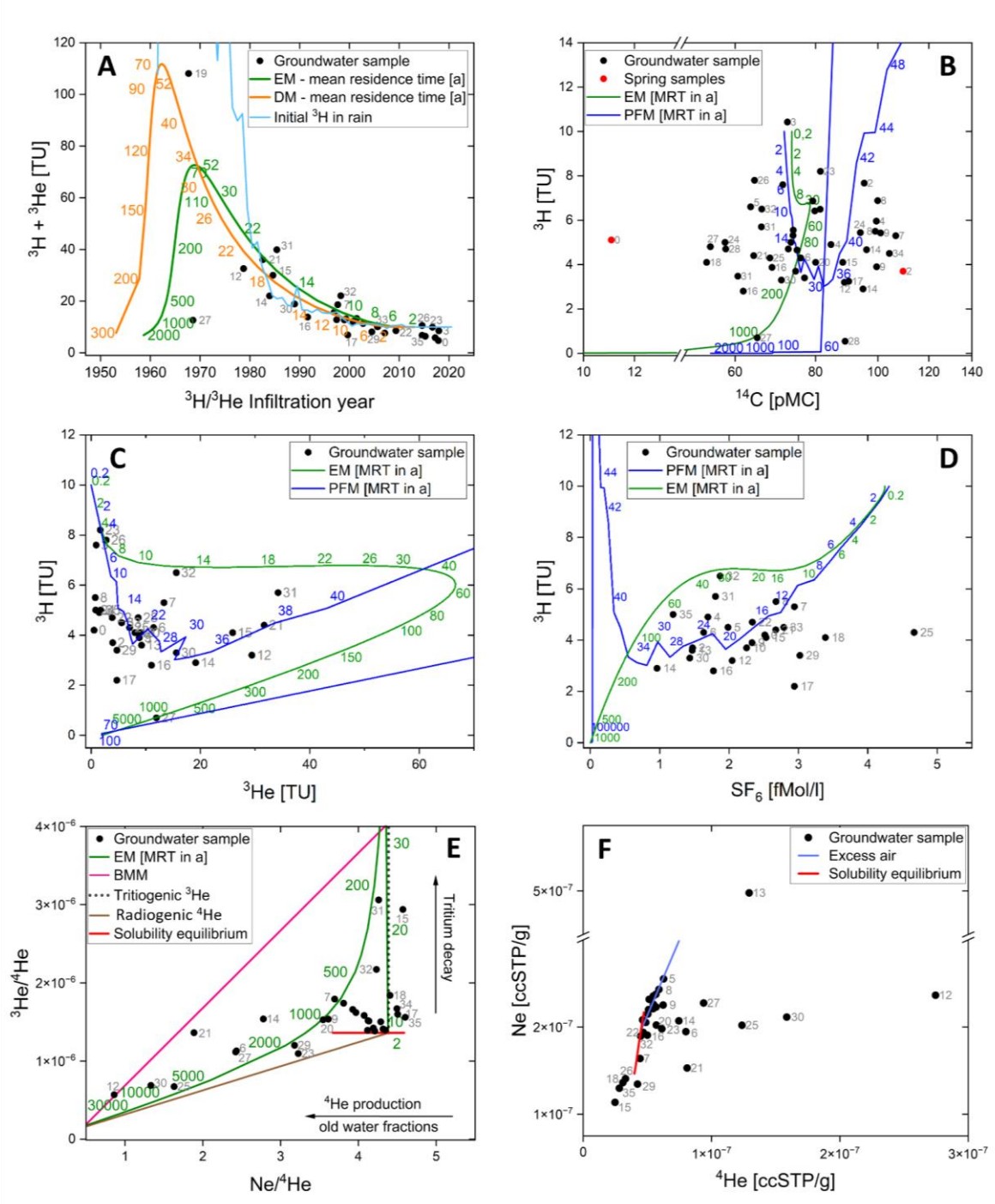

Figure 4. Selected synoptic plots from different environmental tracers using an EM (green), DM (orange), and PFM (blue) as proxies for the aquifer geometry. Groundwater samples are represented by a black point. **A)** initial ($^3$H+$^3$He) tritium versus infiltration year, **B)** tritium versus radiocarbon and springs shown as red points, **C)** tritium versus tritiogenic helium and, **D)** tritium versus sulphur hexafluoride. **E)** shows $^3$He/$^4$He versus Ne/He signatures in sampled groundwater (black points). The green line displays an EM with numbers as mean residence times in years. The pink line shows the binary mixing model (BMM), indicating samples with mixed water fractions above the EM line. Tritiogenic helium as product of tritium decay is shown as a black dotted line and radiogenic helium as decay product of Thorium and Uranium is shown as a brown line. The solubility equilibrium line (red) is running from 0 °C (right end) to 50 °C (left end). **F)** shows Ne versus $^4$He signatures, using the same red solubility equilibrium line as in figure 4E and the excess air line in light blue. Samples to the left below the red line indicate degassed samples. Note y-axis break.

Figures 4C and 4D compare $^3$H with tritiogenic helium and $SF_6$, respectively. Both tracer combinations
indicate infiltration ranges of 4 to 38 years (Fig. 4C) and 12 to 34 years (Fig. 4D). Overall, the samples
display low $^3$H concentrations, averaging 7.5 TU, which is characteristic of young European
groundwater systems (Ying et al., 2024). Sample #27, being at the detection limit in tritium, seems to
have had some tritium originally, since it shows some tritiogenic $^3$He, but this can also be a small
dispersive admixture from the bomb peak. Inspecting the sample cluster #17, #18, #25, #29 of figure
4D shows high $SF_6$ values, for which an air contamination is possible that is also found in figures 4C
and F. Even though some samples show increased concentrations above 3 fMol/l $SF_6$, they are still in a
range that can be explained by atmospheric equilibrium with amounts of excess air uneven in Europe.
No indication for underground production of $SF_6$ was drawn, which is typical for young water (Harnisch
& Eisenhauer, 1998; Friedrich et al., 2013).

Figures 4E and 4F show $^3$He/$^4$He versus Ne/He and Ne versus $^4$He signatures in sampled groundwater.
All samples left below the red solubility equilibrium line (4F) show degassing effects, while sample
#13 shows significant contamination with air. The meeting point of all lines in plot 4E at Ne/He of 4.4
and $^3$He/$^4$He of $1.36 \cdot 10^{-6}$ corresponds to atmospheric equilibrium at 10 °C, while a ratio of $2 \cdot 10^{-8}$ for
$^3$He to $^4$He is assumed for the terrigenic end-member (bottom left end of the brown line). Other
infiltration temperatures correspond to the red solubility equilibrium line between 0 °C (right in Fig. 4E
and up in Fig. 4F) and 50 °C (left in Fig. 4E and down in Fig. 4F). Older water fractions are situated in
areas of low Ne/He ratios and low $^3$He/$^4$He ratios and follow the brown radiogenic helium line. The
decay of tritium delivers only negligible amounts of helium but it is pure $^3$He and results in the vertical
line of tritiogenic helium. The PFM with no mixture would follow the vertical line until all tritium
decayed and then follow the radiogenic line of underground helium production. The EM, which is
displayed only for comparison as a green line, then deviates from these two lines since there is always
a mixture between very old and very young components in an EM. The pink line for the binary mixing
model (BMM) connects a point of 50 TU decayed into $^3$He (top right) to the point of infinite age with
radiogenic helium production (bottom left). Since all samples are located in the space opened up
between the boundaries of the red solubility line, the black tritiogenic line, the brown radiogenic line
and the pink BMM line in figure 4E, they can all be explained by mixed water fractions of recent water
(containing tritiogenic helium) and old water (containing radiogenic helium). It turns out that helium is
the most sensitive qualitative indicator for admixtures of an old component, although this component
cannot be quantified without having the helium concentration in the old end member.

CFC-11 and CFC-12 were determined using the same method as $SF_6$. The plot of CFC-11 versus
CFC-12 (see supplement S3) shows that both species were degraded in the anaerobic sediments of the
aquifers presented here. The results therefore do not provide reliable estimates of groundwater time
scales and are not further discussed.

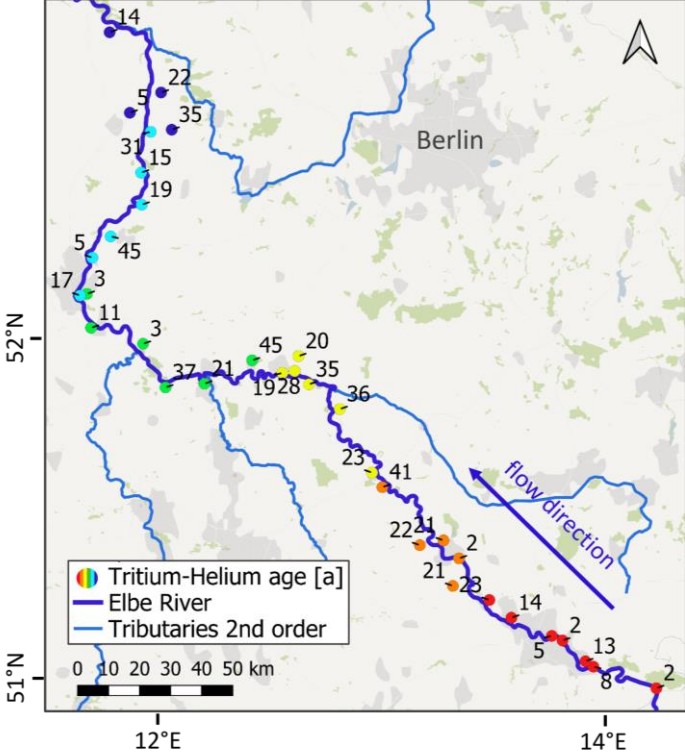

Figure 5. Tritium-Helium age in years of sampled groundwater along a 450 km stretch of the Elbe River. For better spatial orientation: groundwater sampling locations which are located along the start of the studied Elbe section are represented by warm colours. Locations along the end are represented by cold colours. Base Map derived from QGIS desktop.

## 5. Discussion

### 5.1 Stable isotopes

The $\delta^{18}O$ and $\delta^2H$ values of the sampled groundwater (Fig. 3A) indicate minimal evaporation. However, sample #28 shows a more pronounced evaporation of 8‰, which can be attributed to the influence of former gravel pit lakes in the recharge area, likely causing a heavier isotopic signature as lake water mixed with groundwater (Stichler et al., 2008). Similarly, sample #24 exhibits 4.2‰ evaporation, resulting from infiltration through seasonally filled streams such as the former Ehle and Umflutehle (Benettin et al., 2018). The nitrate isotopes $\delta^{18}O$ and $\delta^{15}N$ in $NO_3$ reveal strong enrichment but do not follow uniform enrichment trends (Fig. 3C, D). This heterogeneity can be explained by the diverse aquifer systems and groundwater bodies sampled, which are influenced by agricultural areas under variable land-use practices (Craswell, 2021). The isotopic signatures suggest that organic sources, such as soil organic matter and manure (Clark & Fritz, 2013), dominate the nitrate inputs in most samples. No synthetic fertiliser residues were detected, except in spring samples #1 and #3 within the Saxonian Cretaceous aquifer, which raises questions about potential contamination sources. Denitrification rates across the study area were generally low to moderate, with the highest rates observed in samples #7,

#11, #12 and #20 (Fig. 3D). This partly limited denitrification activity could be attributed to the
relatively young groundwater ages, which may not provide sufficient time for extensive denitrification
processes to occur within the aquifers (Bourke et al., 2019). Similar observations of lower denitrification
in young groundwater have been reported in other European studies (e.g., Rivett et al., 2008; Weyer et
al., 2019). Eh values of the 2022 sampled groundwaters ranging from -236 to +315 mV, while samples
from 2020 show higher values from -42 to +613 mV, indicating more oxidative conditions, making
denitrification less likely.
The $\delta^{13}C$ values of DIC in groundwater (Fig. 3B) reflect the interplay of biogeochemical processes such
as root respiration, microbial activity, carbonate dissolution, and silicate weathering (Mook, 2005). In
this study, $\delta^{13}C$ values range from -23 ‰ V-PDB, characteristic of root respiration from C3 plants,
to - 12 ‰ V-PDB, typical of freshwater carbonate dissolution, highlighting the influence of both
organic and inorganic carbon sources (Meier-Augenstein, 1999). Further, Clark & Fritz (2013) defined
the groundwater $\delta^{13}C$ range from -16 ‰ to -14 ‰ and manure signatures of C are described from -24 ‰
to -16 ‰ (Cravotta, 1997; Vitòria et al., 2004). Both ranges fit our measured $\delta^{13}C$ values. In silicate-
dominated aquifers, DIC levels tend to be lower due to the limited availability of carbonate material for
dissolution (Appelo & Postma, 2005). Chemical weathering of silicates primarily releases cations and
aqueous silica, contributing minimally to inorganic carbon, in contrast to carbonate-rich systems where
calcite           or           dolomite           dissolution           dominates.
The $\delta^{13}C$ values of infiltrating groundwater in agriculture dominated areas (blue points in fig. 3B) are
strongly influenced by soil-derived $CO_2$ from root respiration and microbial decomposition of organic
matter (Gat et al., 2001). Elevated microbial activity in such soils often correlates with higher
denitrification rates (blue points in fig. 3C), further affecting the isotopic composition of DIC. Lighter
$\delta^{13}C$ signatures are therefore typically affected by these agricultural processes and lead to isotopic
enrichment (more positive values; Clark & Fritz, 2013). Another critical aspect are the residence times
within the aquifer, as longer residence times in carbonate-rich aquifers lead to higher DIC
concentrations and $\delta^{13}C$ values closer to the signature of freshwater carbonate minerals. In contrast,
younger groundwater from silicate-rich areas as in our study often shows lower DIC concentrations
dominated by lighter soil $CO_2$ signatures (Clark & Fritz, 2013). Mixing of waters from different flow
paths further explains the variability in DIC levels and carbon isotopic composition.
The sampled springs yielded inconclusive results and were identified as outlier values for various
parameters (shown representatively in Figures 3A, C and 4B). Among others, this might be related to
the inherent complexity of the hosting fractured aquifers, which represent a multicontinuum or at least
dual porosity system possessing no uniform flow path but rather a dynamic mixture of slow and fast
flow paths and accordingly widely spread residence times (Suckow & Gerber, 2022). To gain more
valid insights into such aquifers, a time series of multi tracer (short to long term tracers) measurements
and samples from the upstream source aquifers are necessary (Suckow et al., 2013).
5.2 Environmental age tracers
The finding that only two samples are at the detection limit of tritium is a clear and robust indication
that the catchments investigated are dominated by young water infiltrated after the 1960s. This
indication is the most robust, because tritium is a part of the water molecule itself and therefore not
influenced by gas loss (CFCs, $SF_6$, $^3He$) or geochemical processes (CFCs, $^{14}C$). Table 1 presents further
indicative age estimates which are derived from figures 4 and 5. Although these estimates are
approximations, they are in agreement with and refine the results of Wendland et al. (2004), who
estimated the mean residence time in the Elbe catchment area to be 25 years using a stochastic model
with large uncertainties. However, Esser (1980) underestimated the age distribution by stating that water
fractions older than 10 years account for only a small proportion of about 5 %.
Table 1: Summary of calculated and modelled groundwater time scales from multiple tracers. *Without
outlier values > 45 years.

| Tracer system | Used approximation | Age estimates [a] | Figure |
|---|---|---|---|
| $^3H/^3He$ age | Equation 1 | 2 - 41* | 5 |
| $^3H+^3He$ vs. infiltration year | DM, EM | 0 - 20 | 4A |
| $^3H$ vs. $^3He$ | PFM | 4 - 38 | 4C |
| $^3H$ vs. $^{14}C$ | PFM | 0 - 42 | 4B |
| $^3H$ vs. $SF_6$ | PFM | 12 - 34 | 4D |


Although the majority of the samples exhibit "young ages", several exceptions are noteworthy. Samples
#12, #17, #19, and #27 display evidence of admixture with older, tritium-free waters, as they fall below
the $^3H$ line for precipitation (light blue, Fig. 4A). Further samples #18, #24, #27, #28 show admixtures
of older water fractions according to their $^{14}C$ values around 60 pMC (Fig. 4B) and more evident,
according to their high $^4He$ values for sample #12, #21, #25, #30 (Figs. 4E and 4F). These components
must have infiltrated prior to the first nuclear tests in 1953, as $^3H$ concentrations in precipitation have
been significantly elevated since that time due to atmospheric nuclear testing (Schmidt et al., 2020).
Additional samples (#06, #12, #14, #21, #25, #27, #30) also show clear admixtures of older waters
according to $^3He/^4He$ vs Ne/He data in figure 4E. These fall to the left of the EM (green line, fig. 4E)
and the state of equilibrium (red line, fig. 4E) and show correspondingly high radiogenic $^4He$, produced
from the decay of U and Th in the aquifer. This suggests a mixing of old and young waters, as helium
is the most robust indicator of old groundwater admixed to young water (IAEA, 2013).

Sample #19 exhibits an exceptionally high tritiogenic $^3$He value of 103 TU, significantly exceeding the average $^3$He concentration of 8 TU observed across the other samples, while maintaining a normal $^3$H concentration of 5.7 TU (Fig. 4). A plausible explanation would be that this is nearly unmixed water from around the thermonuclear bomb peak. Some samples (e.g. #7, #15, #21, #31, #32) exhibit $^3$H concentrations exceeding those of precipitation and thus the natural geogenic background (Fig 4A, light blue line). This suggests these groundwater samples have been influenced by bank filtrate or influent conditions in general, as the Elbe River is highly enriched in $^3$H with 50 - 100 TU and more (Schubert et al., 2020; Zill et al., 2023). This is due to the cooling water release of the Czech nuclear power plant into the Moldau River (Hanslík et al., 2017), the largest tributary of the Elbe. Sampling wells close to the Elbe River and situated in the floodplain are therefore marked with an asterisk in supplementary material S1 and S2, suggesting higher $^3$H concentrations than the natural background.

Several samples (#15, #18, #26, #29, #35) display evidence of either geogenic degassing or degassing during sampling (see Fig. 4F, below the red solubility line). These degassing processes render the samples unsuitable for noble gas (Ne/He) analyses. A geogenic mechanism for noble gas degassing could involve nitrate reduction processes in nitrate-rich groundwater (e.g., #26 and #35 showing $NO_3$ concentrations of 72 and 154 mg/l, respectively), generating $N_2O$ or $N_2$ bubbles. These bubbles equilibrate with dissolved noble gases, effectively partitioning them into the gas phase and reducing their concentrations in the groundwater (Kipfer et al., 2002). Sampling-related degassing may occur when air bubbles form due to de-pressurisation during sampling, leading to stripping of He and Ne. The $^{14}$C analysis of DIC confirms the young age of groundwater along the Elbe River (Fig. 4B). As values down to 60 pMC can be readily explained by geochemical interactions, only one sample (#1) shows an unequivocal indication of radioactive decay. All other measured $^{14}$C values in combination with tritium indicate recent recharge or geochemical interactions of the carbonate system. Additionally, the generally low radiogenic $^4$He concentrations observed in the samples are typical for young groundwater in Europe, which has limited time to accumulate helium from radiogenic sources (Broers et l., 2021; Desens et al., 2023).

## 6. Conclusions

Our study determined the time scales of groundwater flow in Cretaceous and Quaternary aquifers along a 450 km stretch of the German Elbe River. Using the environmental tracers $^3$H/He, $SF_6$, CFC's and $^{14}$C and a lumped parameter approach, flow times in the range of a few decades were derived. In general, this is a young mixed groundwater system that can react quickly to changes, while the admixture of older groundwater is detectable in smaller proportions, except for multi-component samples like #12, #18, #21, #27, #29, #30, #31. Spatially, this refers to river sections of stream km 150, 218, 280, 350-

370 and 395. No valid statement could be made about the age of spring water in the mountainous upstream areas. Natural nitrate degradation is low to moderate with highest nutrient concentrations of $NO_3$ and $PO_4$ in samples #6, #8, #12, #26, #35 and #10, #32x, #34, respectively. In the coming decades, a reduction in groundwater-borne nutrient concentrations entering the Elbe River is anticipated. Consequently, the impact of groundwater discharge on riverine eutrophication is expected to be less significant in near-future scenarios. This information is crucial as it has been shown that groundwater can contribute significantly to riverine eutrophication in the Elbe, and therefore adapted management to mitigate algal blooms and protect aquatic ecosystems is valuable. In the future, the additional use of i) the age tracer $^{39}$Ar would be beneficial, as it covers the age spectra between the tracers $^3$H and $^{14}$C used here and ii) wells with filter screens less than 2 m should be preferentially sampled, as they result in minimally affected tritium-helium ages.

## Author contribution
**JZ**: Data curation, Formal analysis, Funding acquisition, Project administration, Software, Validation, Visualization, Writing – original draft; **ASu**: Formal Analysis, Software, Methodology, Writing – original draft; **UM**: Supervision, Conceptualization, Writing – original draft; **JS**: Formal analysis, Resources, Writing – original draft. **ASch:** Formal analysis, Resources, Writing - original draft. **CS**: Project planning, Methodology, Supervision, Conceptualization, Resources acquisition, Writing – original draft;

## Declaration of Competing Interest
The authors declare that they have no known competing financial interests or personal relationships that could have appeared to influence the work reported in this paper.

## Data availability
All data are given in the supplement.

## Acknowledgement
The authors thank the DBU for funding the scholarship of the first author. We are grateful to the Dept. of Catchment Hydrology for financially supporting the study and Ronald Krieg, Ralf Merz, Stefan Geyer and Kay Knöller for field assistance. We thank Silke Köhler, Gabriele Stams, Wolfgang Städter, Stoyanka Schumann, Gudrun Schäfer, Stephan Weise and Marion Martienssen for analytical support. We are extremely grateful to Georg Houben from the Federal Institute for Geosciences and Natural Resources for covering part of the costs of the helium analyses.

## Financial support
The study was granted by Deutsche Bundesstiftung Umwelt (DBU) through the scholarship to JZ [grant number 20019/637]. The article processing charges for this open-access publication were covered by the Research Centre of the Helmholtz Association through project DEAL. Parts of the analytical costs have been covered by the Federal Institute for Geosciences and Natural Resources, department of groundwater quality and protection.

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
