# Peer review of "Will groundwater-borne nutrients affect river eutrophication in the future? A multi-tracer study along the Elbe River"

_EGUsphere, 2025_

## Author Response (AR1)

Response to editor and reviewers

Title: Will groundwater-borne nutrients affect river eutrophication in the future? A multi-tracer study along the Elbe River

Journal: Hydrology and Earth System Science

Manuscript Number: **egusphere-2025-642**

julia.zill@ufz.de

**Editor Nunzio Romano**

*Dear Authors, Two experts reviewed your submitted paper and gave it similar ratings. The referees also provided useful comments during the discussion phase of the journal, and are willing to evaluate the changes where appropriate. Please submit the revised version of your study together with the point-by-point replies to all comments received.*
Answer: Dear Editor, thank you very much for the smooth review process and your feedback! We discussed the reviewers' comments and provided herewith a point-by-point answer.

**Reviewer #1** (https://doi.org/10.5194/egusphere-2025-642-RC1)

*The paper addresses a complex set of issues in relation to assessing the current and future risk levels of ongoing groundwater borne contaminants entering the Elbe. In addition, a suite of environmental tracers and LPM techniques are brought to bear on the issue. This is not to mention the challenge of applying these methods (and logistics) to a 450km stretch of a major European river suffering eutrophication.*
*In the context of two previous papers, Zill et al make a highly creditable effort to arrive at their conclusions which contribute significantly to overall future management of the Elbe. The paper will be of interest to readers of the HESS journal. Subject to the authors' consideration of the comments above and those in the annotated pdf, the paper should be accepted for publication.*
Answer: Thank you for your positive and supportive feedback, which significantly contributed to the improvement of the paper!

*The paper title needs reviewing. in so far as it needs to specify the River Elbe. The conclusion falls rather short of stating a clear answer to the question posed in the title. The answers are there in the paper, but they need to be emphasised more strongly since the authors pose the question.*
Answer: We changed the title into ''Will groundwater-borne nutrients affect river eutrophication in the future? A multi-tracer study along the Elbe River''. And we changed the conclusion section accordingly.

*Line 60. It would strengthen this paper to have at least one or two orthogonal transects to the Elbe selected at key locations and apply a 2D vertical section solute transport code that corroborates transport rates, fluxes and loads.*
Answer: Although the suggestion is very interesting and doubtless a worthwhile extension, it exceeds the original concept of the study and its scope. The transport of solutes, rates and fluxes are generally proven and specifically dependent on the time of observation. We here present an integral view on the groundwater age distributions. Since the approaching groundwater is a mix of different old groundwater, we do not try to determine a precise "age" but estimate the most probable MRT. To tackle the question whether or not, groundwater from older periods (from before 1990), when

fertilizing schemes have been different compared to after that year, are still on the way. However, since the study is also hosted under a larger umbrella, modelling the connectivity of the surrounding aquifer bodies to the river network including the Elbe is an ongoing task and will be presented at a later stage.

*Paragraph 105. It would be helpful to further expand the relationship of his paper to those of Zill 2023, 2024, i.e. how the multitracer approach is a follow-on from these earlier papers, i.e. completing more-or-less a trilogy. Such emphasis would make the context of this paper a lot clearer.*
Answer: An expanded explanation of the two previous papers is mentioned in section 2. "Study Area" within the lines 121 - 134, page 4.

*Some lapses in English and English meaning occurred (e.g. line 155-165 ) that need to be improved.*
Answer: The wording of this section was improved. Lines 159 - 168, page 5.

*Many markup notations are in the attached paper pdf file.*
Answer: Thank you for the many hints and improvements in that file. We have implemented the suggestions accordingly.

**Reviewer #2 Matthew Currell** (https://doi.org/10.5194/egusphere-2025-642-RC2)

*This paper by Zill et al make a nice contribution to knowledge of tracer-based groundwater residence time estimation, in the context of nutrient transport to major river systems. The data are high quality and the interpretations are largely sound. A few suggestions which I hope will improve the quality of the paper prior to publication.*
Answer: Thank you Matthew for your constructive feedback, which very much improved the paper.

*Line 45 - 50: You may be 'over promising' here. It is very hard to resolve the history of nutrient input with sufficient temporal resolution to match the loads to specific policy initiatives and their time-periods (and the paper does not go to this level of detail). Perhaps de-emphasise this (or reality check it), so readers aren't disappointed with the final outcome.*
Answer: Yes, indeed the passage is formulated as if it is just to be determined. We shared your point and changed the sentence accordingly to relax the "promise". Lines 50 - 57, page 2.

*Line 56 - Model calibration is not really an objective in itself, and further, it is very difficult to achieve using residence time-tracers (which themselves require model assumptions to be made prior to mean residence time estimation). The later parts of this section (lines 60 onward) are to me the more important aspects and where the highest value of the approach lies.*
Answer: We share your point that calibration is not a pure objective, however, validation is a goal in itself. We delete the calibration aspect. Line 59, page 2.

*Line 90 - Note clear what you mean by 'all nutrients are retained'. Retained where? What does this have to do with diffusion/dispersion?*
Answer: We improved the formulation and changed this to ''All nutrients in the aquifer are subject to dispersion and diffusion effects.'' Line 94, page 3.

*Line 127-128: Not clear whether this is a surface or sub-surface pathway for nutrients to the river (clarify).*
Answer: Field drainages are intended to dewater the agricultural fields. It is a piping network buried about 1 m below ground which efficiently transports soil water towards the Elbe River. They are installed in variable but shallow depths but subsurface. We clarified this in the text. Line 132, page 4.

*Line 130: Disorientation of the chemistry: meaning is not clear*

Answer: That is a typo, deterioration of the water quality was meant. We reword this sentence. Lines 133, page 4.

*Line 150-151: Vulnerability to contamination?*

Answer: Yes, correctly. In our case that term is used to express quality hazards, but no quantitative issues. Lines 153, page 5.

*Line 165: How were the 'best' models selected, or were all three used to assess all the samples? Best to explain this here and/or in the relevant section of the methods.*

Answer: All of the three models have been applied and results were analysed in terms of "best fit". By comparison, we selected the best model for our data set. We changed the wording to prevent the word "fit", since we compared the results and did not fit our data. Lines 167-169, page 5.

*Line 195: Do you mean during and after this period?*

Answer: We believe our formulation is clear: We need to define the initial concentration during the recharge and the dilution during subsurface flow by "dead" carbon due to dissolution of million years old limestone rocks. So, both periods are meant. Lines 202 - 203, page 6.

*Line 343: The %evaporation estimates depend on relative humidity (I believe). How was this estimated or what value used to calculate the enrichment according to the evaporation line?*

Answer: We applied 60 % humidity and 20 °C, which are reasonable values for the area. We added this information in the text. Lines 348-349, page 11.

*Line 349: Should be 0 to 6.7 mMol/l (given the maximum quoted in the next line)*

Answer: We improve this line. Line 355, page 11

*Line 351: Capital letter for Cretaceous*

Answer: We improve this line. Line 359, page 11

*Line 354: Suggesting*

Answer: We improve this line. Line 362, page 11

*Line 381: Was this a deeper well than the others? (the 3H free sample?)*

Answer: The well was slightly deeper than the others, but the base of the screen was 23 m below ground only, so not really a depth, where we assume tritium free water yet.

*Figure 4: Good figure, though I am not quite clear why the blue line has this shape - with the youngest water at ~10TU and ~70pMC. Is this considered the starting point for current-day recharge and if so why is pMC 70 (geochemical reactions?)*

Answer: The value 70 pMC is a factor applied to the 14C input function. It is based on literature from Hua et al., 2022 (Hua, Q., Turnbull, J. C., Santos, G. M., Rakowski, A. Z., Ancapichún, S., De Pol-Holz, R., et al. (2022): Atmospheric Radiocarbon for the Period 1950–2019. *Radiocarbon, 64*(4), 723-745). Youngest Water at 10TU comes from the Vienna input curve without correcting factor (IAEA/WMO, (2016): *Global Network of Isotopes in Precipitation. The GNIP Database*).

*Discussion: This section is quite long and repeats information from the results section. Try to break up the long paragraphs and remove the repetition. You don't need to discuss all data types, just stick with those that allow you to draw and support the main conclusions.*

Answer: We split the long discussion sections and shorten the repetitions wherever possible.

*Line 470-474: This repeats earlier text in the results*

Answer: This section was shortened. Lines 479-482, page 16.

*Line 523: 1960s*

Answer: We improve this line. Line 529, page 18.

*Line 555: This was mentioned above already. Don't repeat it (cut the earlier text if this works better)*

Answer: The earlier section was shortened. Lines 388-390, page 13.

*Line 549 to 573: A very long paragraph. Better to break it up.*

Answer: We split this paragraph into two. Lines 552-577, page 19.

*Line 563: Or N2O gas?*

Answer: This is in general possible, when the process of denitrification is not complete. To generate N2, the interim product N2O is developed and if the process is interrupted or terminated, it will not be further reduced to N2. We added N2O in the text. Line 567, page 19.

*Overall conclusions/findings: I think the beginning of the article discussed the relevance of the research to assessing nutrient loads to the Elbe River, and constraining how these have been changing through time. In the end, there wasn't a lot of discussion about this topic based on the data. Are there waters from certain regions or with particular residence times that were associated with more or less nutrient load, and if so, can you say anything about the times/locations where the river may be most vulnerable to input of contamination from the sub-surface? This would give your study more relevance and applicability.*

Answer: We improved the conclusions paragraph by including a respective section to outline areas of highest risk of pollution. We included locations with oldest groundwaters and those, where the highest load of nutrients is determined in the groundwater samples. Lines 585 - 592, page 20.